# Study of Wettability of Polyethylene Membranes for Food Packaging

Sebastiano Vasi [1,*], Giovanni Ceccio [2,*], Antonino Cannavò [2], Pavel Pleskunov [3] and Jiří Vacík [2]

1 MIFT Department, University of Messina, Viale Ferdinando Stagno d'Alcontres 31, 98166 Messina, Italy
2 Department of Neutron Physics, Nuclear Physics Institute (NPI) of the Czech Academy of Sciences (CAS), 250 68 Husinec-Rez, Czech Republic; cannavo@ujf.cas.cz (A.C.); vacik@ujf.cas.cz (J.V.)
3 Department of Macromolecular Physics, Faculty of Mathematics and Physics, Charles University, V Holesovickach 2, 180 00 Prague, Czech Republic; pleskunov@kmf.troja.mff.cuni.cz
* Correspondence: sebastiano.vasi@unime.it (S.V.); ceccio@ujf.cas.cz (G.C.)

**Abstract:** In this study, the wettability of PET membranes (prepared with different pore sizes) treated by UV irradiation, thermal annealing or doping with metal nanoparticles was investigated. The wettability was studied using the contact angle method based on the optical microscopy. The membranes were analyzed before and after pore etching, and after each applied treatment. It turned out that membranes with different pore sizes exhibit different wetting behavior. Of particular interest are membranes with 0.53 μm pores. When pristine, they show high hydrophobicity (a high contact angle), but after treatment (some of which can be considered as an accelerated aging), their wetting characteristics swap between a hydrophobic and hydrophilic state. Interactions between packaging material and food and the external environment through fine control of wettability could have a major impact on maintaining product quality.

**Keywords:** wettability; polymers; membrane; food packaging; aging

## 1. Introduction

An important and unmissable aspect of plastic materials (polymers) is that they cannot be replaced by other materials in specific industrial and technological areas, such as electronics (as separators in lithium-ion batteries, flexible electrodes, liquid-crystal displays) [1,2], medicine (e.g., antimicrobial membranes) [3], or modern environmental technologies (such as micro-filtration and water purification) [4]. In addition, plastics of different kinds are also particularly suitable for food (and beverage) packaging [5], as they allow for easy and efficient conservation of fast-spoiled food and food ingredients, and are also refillable several times. For this purpose, the packaging material should have certain properties, such as non-toxicity, flexibility, transparency (or opacity), and chemical and thermal resistance. It turns out that polymers of various types meet such properties and are thus suitable (sometimes unique) materials for food packaging [6]. For instance, PolyEthylene Terephthalate (PET) is one of the most used synthetic polymers in the food packaging industry due to its low-cost production and good performance. PET is resistant to moisture and microbial, enzymatic, and chemical reactions. It is a strong, lightweight material with excellent long-life properties, enabling its reuse (in fact, it is the most widely recycled plastic [7–9]). In addition, PET requires fewer admixtures (compared to other plastics) to be packaged. Importantly, PET does not change food when in contact with it [10]. In food packaging, key aspects to consider are the food quality, freshness and the food safety, all of which depend on the packaging property and quality. Food production requires a significant amount of resources, and by preserving food quality for as long as possible, the possibility for wasting food is reduced. Therefore, improvements on the properties and quality of packaging materials can result in benefits in the food system.

In food packaging, the tendency of foodstuff to interact with packaging is one of the key factors that affect the preservation of the food itself [11]. The adherence of the food residues to the packaging may enhance oxidation, deterioration and consecutive bad flavor, thus increasing waste and resulting in lowering of overall product quality. In this respect, most packaging films are made to be hydrophobic and are surface-treated to improve their overall resistance to chemical interactions with foodstuff. In addition, it is important to understand how ageing affects the plastic film parameters, e.g., how the plastic film will interact with liquid over time under certain environmental conditions. The interaction between the surface of the packaging and the wetting liquid depends on the intermolecular forces involved [12]. The contact angle method can be used to investigate such surface interactions and to determine the wettability of the plastic surface (i.e., the ability of liquid to maintain the balance between the intermolecular interactions of adhesive and cohesive types on the plastic). With increasing contact angle, the interaction between the surface and the liquid decreases [13].

The main goal of this work is to uncover possible applications of the ion-track etching technique for membrane preparation to be used for food packaging, by controlling the hydrophobicity of the investigating films. In fact, the ability to control hydrophobic or hydrophilic behavior may have a strong impact on food packaging, reducing the amount of water that lie of the surface of packing films. Achieving such results would improve the functionality of the PET packaging in a way that can ensure food freshness, quality and safety for longer, and in turn, lower the food spoilage rate. This investigation was carried out by means of contact angle measurements of PET membranes with etched pores of different sizes and area densities. In addition, the modification of PET (induced by etching and formation of pores) makes is possible to determine the effect of (accelerated) ageing on the wetting properties of PET. Innovation regarding practical applications of membranes produced by the ion-track etching technique and related studies on their wettability in terms of food packaging, up to now, as far as we know, have been rarely investigated.

## 2. Materials and Methods

The membrane samples were prepared using PET (Hostaphan® Mitsubishi Polyester Films) foils (irradiated in JINR Dubna by Dr. P.Y. Apel), in which pores of 0.53 μm, 2.4 μm and 7.0 μm in diameter were formed by ion-track etching (with a track density of $10^8/cm^2$ for the foil with 0.53 μm pores and $10^6/cm^2$ for the foils with the 2.4 μm and 7.0 μm pores) [14–16]. The pore diameters were analyzed by SEM (JSM-7200F, Jeol Ltd., Tokyo, Japan). The membranes were then divided into five groups, each group being treated differently: group A was annealed at 100 °C for 24 h in air; group B was exposed to 365 nm UV irradiation (Herolab GmbH Laborgerate, Wiesloch, Germany) for 24 h with the intensity of 1200 μW/cm² (this can be considered as an effect of accelerated aging); the last three groups of membranes (C, D, E) were doped with a suspension of nanoparticles (NPs) (all from NNCrystal US Corporation, Fayetteville, AR, USA)—one set with 0.1 mL of a solution containing 0.05 mg/mL of 10 nm Ag NPs (C), one set with 0.1 mL of a solution containing 0.05 mg/mL of 10 nm Au NPs (D) and one set with 0.1 mL of a solution containing 0.05 mg/mL 5 nm Au NPs (E). After doping, the membranes were dried in a controlled environment in standard room conditions The modified membranes (together with the non-treated sample and a pristine foil) were investigated for wettability by a trinocular microscope (Olympus BX41M) arranged in a horizontal configuration for contact angle analysis. The contact angle is the angle at the interface where liquid (generally a drop of water), air, and solid (the investigated surface) meet, and its value is an estimate of how likely the surface is to be wetted by water [17].

In particular, if we have water strongly attracted to the hydrophilic solid surface, the contact angle will tend to be 0° and a water droplet will stretch on the film surface. In contrast, less hydrophilic surfaces are characterized by a contact angle up to 90°, which exceeds this value when the film is hydrophobic. On these types of surfaces, the droplets are attached to the film with the smallest possible surface. In this case, the contact angle provides information about the free energy of the surface when there is a direct interaction

between the surface and the liquid. For our measurements, a droplet (1 μL) of distilled water was applied to the PET film using a calibrated micro-syringe. The contact angle measurements were performed directly using a high-resolution CCD camera (Optika microscopes, model C-P8) aligned with an optical microscope for image recording. During the measurement, the sample was placed in a movable holder in front of the optical microscope. A 10× magnification was applied for the contact angle analysis. Measurements were repeated 6 times for each sample, and the contact angle was determined as the mean value of all 6 measurements.

## 3. Results

Figure 1 shows SEM pictures of the prepared nuclear PET membrane after the latent-track etching. One can see that the pores are well-defined, so the average pore diameter could be determined for each membrane. In Figure 1a, where the pores are well-separated from each other, the average pore diameter was found to be 7.0 μm. Figure 1b illustrates pores with an average diameter of 2.4 μm and a distance between each other of about 1 μm. As reported in Figure 1c, the smallest pores, with an average diameter of 0.53 μm, are relatively densely packed together, so some of them are merged. Before the measurement of the contact angle between the water and modified PET membranes, pristine foil and untreated membranes were analyzed (we identified the pore-free and non-treated PET foil sample with the name "pristine". Instead we called "no-treated" the PET foil only with pores but without treatment). For example, Figure 2 illustrates a water droplet attached to the pristine PET foil. In this case, the contact angle was found to be 68°, indicating the initial more hydrophilic behavior of the untreated plastic. On the pristine porous membranes, the contact angles were measured to be 99.4°, 58.3° and 0° for the 0.53 μm, 2.4 μm and 7.0 μm pores, respectively.

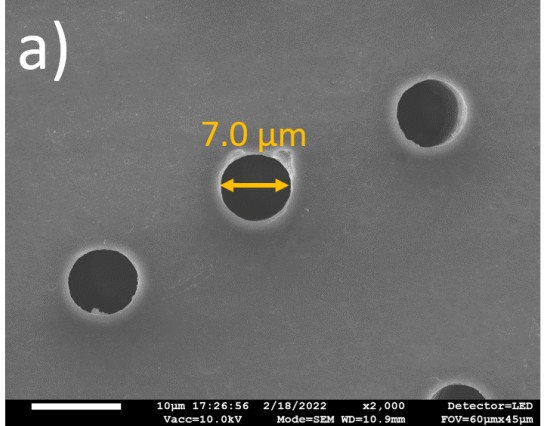
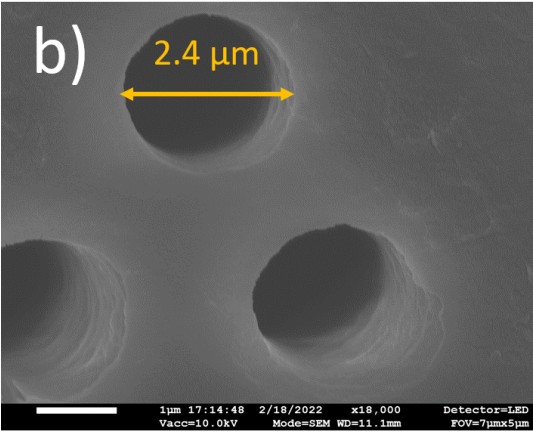

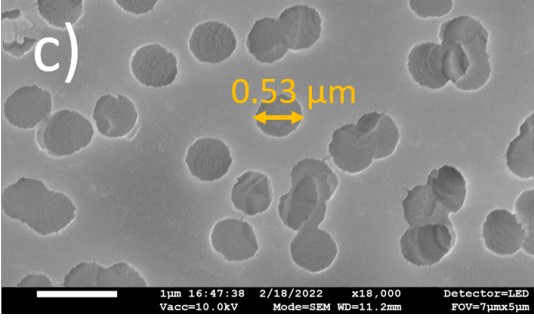

**Figure 1.** SEM of the PET membranes before treatment: (**a**) a membrane with the largest pores of 7.0 μm, (**b**) membrane with average pores of 2.4 μm and (**c**) a membrane with the smallest pores of 0.53 μm.

The subsequent treatment, which can be carried through annealing, UV irradiation and doping, led to a change in the wettability depending on the pore size and membrane treatment. In Figure 3, the contact angle of the water droplet with the 7.0 μm pore membrane after annealing and UV irradiation is shown. One can see how the droplet of water reacts to the surface: after annealing, the membrane became less hydrophilic (than untreated membrane)—the contact angle was 80.1° (Figure 3a). After UV irradiation, this was only partially the case—the contact angle was 48.3° (Figure 3b).

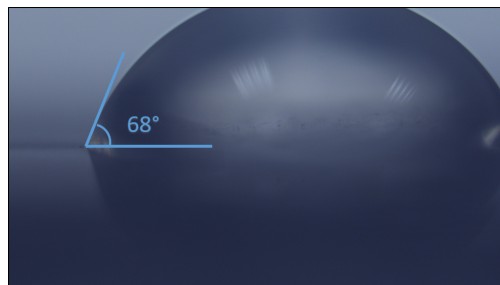

**Figure 2.** Optical image of the contact angle between the pristine PET and the droplet of distilled water.

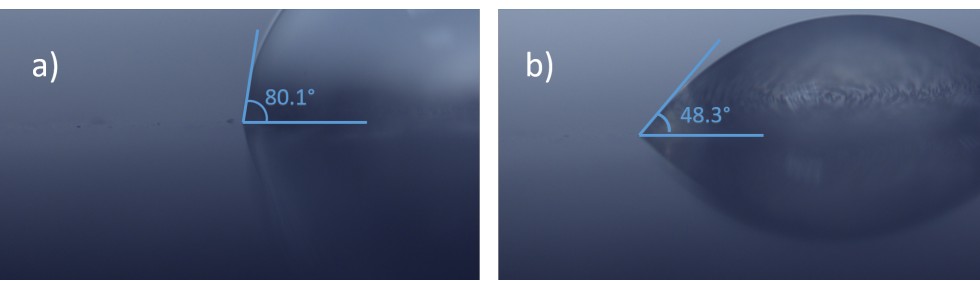

**Figure 3.** Optical image of the contact angle between the water droplet and the PET membrane with 7.0 μm pores after thermal annealing (**a**) and UV irradiation (**b**).

Figure 4 represents the average contact angle measured on membranes (with 0.53 μm, 2.4 μm, and 7.0 μm pores) for all different treatments (annealing, UV irradiation, NPs doping). As shown in this figure, the contact angle depends not only on the pore diameter but also on the type of treatment. For instance, the membranes with 2.4 μm pores show strong hydrophilic behavior after all treatments—the contact angle is 0° (and even in the pristine state, the contact angle is lower than for pure PET). The untreated membranes with 7.0 μm pores are strongly hydrophilic, but after UV irradiation and annealing they become partially hydrophobic (the contact angle after annealing becomes even higher than for pure PET).

As illustrated in Figure 4, an interesting wetting behavior of the membranes appeared after doping with NPs. In the case of membranes with 7.0 μm and 2.4 μm pore diameters, these membranes became fully hydrophilic (with the contact angle 0°). However, for membranes with a pore size of 0.53 μm, the contact angle decreased, but it always retained some hydrophobic features. This is probably due to the short-range forces created between the carboxyl groups and the metallic NPs penetrating the pores and being trapped in large numbers at the pores' entrance. Figure 5 again demonstrates the water droplets on the membrane with the 0.53 μm pores before and after thermal annealing. The contact angle measurement shows that after annealing it dropped by about 7%, as can also be seen in Figure 4.

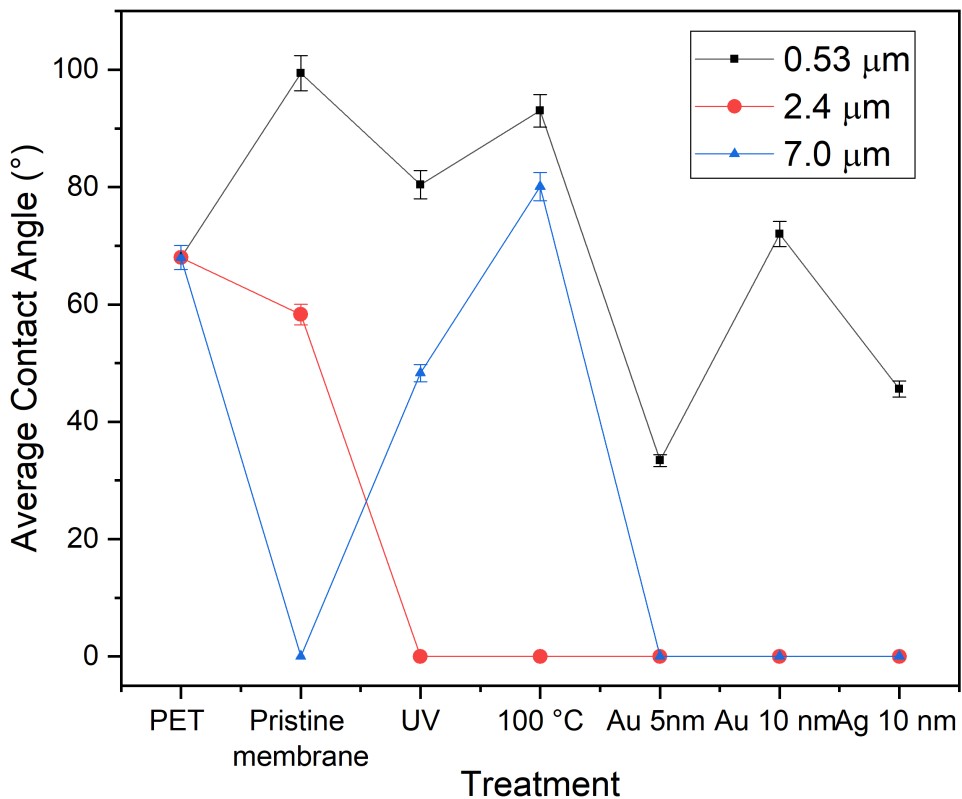

**Figure 4.** Contact angles for PET membranes with different pore diameters before and after treatments (UV irradiation, thermal annealing and doping with metallic NPs).

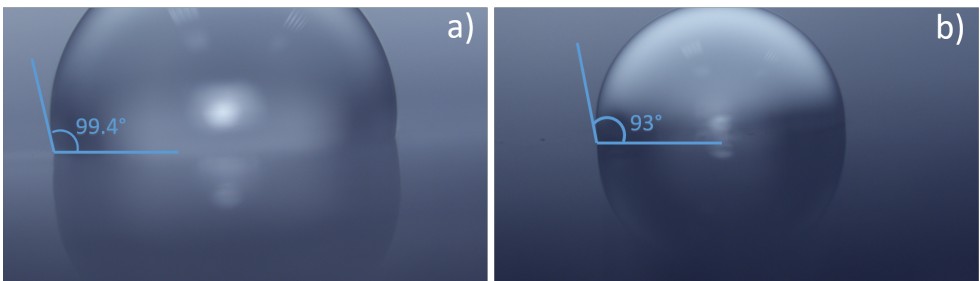

**Figure 5.** Optical image of the contact angle between the water droplet and the PET membranes with pores of 0.53 μm diameter before (**a**) and after (**b**) thermal annealing.

## 4. Discussions

In this experiment, polymeric membranes were produced with different pore sizes by the ion-track etching technique, and their wettability was investigated by means of contact angle measurements. The wettability—and consequently hydrophobicity—is an important aspect to be considered when a material should be used for food packaging, because the tendency of foods to interact with the packing affects the durability of the food itself. For this reason, it is highly desirable for protective films to be hydrophobic in order to avoid accumulation of water in contact with food. The investigated membrane shows a dependence of wettability on the pore dimension. It is well-known that when reducing the pore size, the wettability decreases, creating a highly hydrophobic surface [18,19]. In particular, it was observed that pristine membranes with 0.53 μm pores have a very high contact angle (higher than pure), as expected [20]. The applied treatments on these membranes reduced the contact angle. Annealing on the polymeric membrane induces slight changes in the pores' geometry and aperture. Meanwhile, UV may affect the free radical (formed during etching on the pore walls [21]) and induce changes in the

electrostatic fields on the pores walls, in particular, close to the opening. These effects change the interaction of the water molecules with the surface of samples. Upon investigation, the membrane with 2.4 μm pores appears unsuitable for food packing, because it shows a low contact angle not only in its pristine state, but also after the treatment (that, as mentioned, can be considered as ageing parameters during the life-time of PET films), becoming strongly hydrophilic. If used for food packaging, after exposure to sunlight or if placed in a warm environment, this membrane can start to accumulate water from moisture or from the food itself, creating a dangerous environment that will lead to degradation. The last analyzed membrane with 7.0 μm pores shows an opposite response than the 2.4 μm pores. The initial hydrophilic tendency changed after UV and annealing treatment. After UV processing, the contact angles increased slightly, but after annealing it became almost hydrophobic. This is not a favorable condition for food packing, but shows high potentiality of the technique for the manipulation of the wettability of the membranes, in order to give them the needed characteristics for practical application. In addition to the aforementioned treatments, the effects due to nanoparticles' coating were investigated. For the biggest pores sizes, 2.4 and 7.0 μm, the films were hydrophilic with a 0° contact angle, but for the smallest pores, the contact angle is not equal to zero. The hydrophobicity of the membrane is not as high as in the other condition, but this can open new investigations into the possibility of improving the hydrophobicity of such a membrane in presence of nanoparticles. This may be of interest for the addition of substances with potential antimicrobial activity in order to ensure greater food safety and quality [22]. For example, Ag NPs can be used for preventing the formation of fungi and bacteria (thanks to the biological properties of Ag) [23], and this can improve their use as food and beverage packaging. Or, is also possible to investigate the possibility of using hydrophilic and hydrophobic nanoparticles or coatings [24–27].

## 5. Conclusions

In this paper, we show the feasibility of using a polymeric membrane of PET, prepared by ion-track etching, for food packaging. The main advantages are the possibility to select a more suitable pore shape and subsequently manipulate wettability in order to match the desired condition. In particular, it was shown how the membrane with the largest pores passed changes from hydrophilic to hydrophobic using external stimuli. These results make an important contribution to food safety and quality by improving PET packaging properties, a widely used food-contact material. This prolongs food quality and freshness (thus reducing wastage), and most importantly it ensures food safety. Another interesting feature found was the possibility to keep the hydrophobicity high by introducing nanoparticles, and this can open new and interesting possibilities due to the antibacterial properties of some nanoparticles (for instance Ag). This aspect can be further investigated in future works.

**Author Contributions:** Conceptualization, S.V. and G.C.; methodology, G.C. and S.V.; validation, S.V. and G.C.; investigation, G.C. and P.P.; resources, J.V.; data curation, S.V. and A.C.; writing—original draft preparation, S.V. and G.C.; writing—review and editing, J.V. and A.C.; funding acquisition, J.V.; All authors have read and agreed to the published version of the manuscript.

**Funding:** The project was supported by the Grant Agency of the Czech Republic (Grantová Agentura České Republiky), project No. 22-17346S.

**Institutional Review Board Statement:** Not applicable.

**Data Availability Statement:** All the data presented in this study is already available in the manuscript itself.

**Conflicts of Interest:** The authors declare no conflict of interest.

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
