# Peer review of "Study of Wettability of Polyethylene Membranes for Food Packaging"

_sustainability, doi:10.3390/su14105863_

Round 1

Reviewer 1 Report

Please see the following comments.

11 Quotations:

 p 2 in sustainability-1656220-peer-review-v1 (1)

10 nm Au NPs (D) and one set with 0.1 ml of a solution containing 0.05 mg/mL 5 nm 67 Au NPs (E)

Comment: 

what is difference between AuNP (D) and AuNp (E)? is it only 10 nm vs. 5nm

p 2 in sustainability-1656220-peer-review-v1 (1)

(Hostaphan)

Comment: 

what is Hostaphan?

p 2 in sustainability-1656220-peer-review-v1 (1)

ion-track etching

Comment: 

Please provide the details of PET membranes with different pore sizes. Also what is the distribution of pore sizes

p 2 in sustainability-1656220-peer-review-v1 (1)

(C, D, E) were doped 64 with a suspension of nanoparticles (NPs)

Comment: 

how were membranes dried?

p 2 in sustainability-1656220-peer-review-v1 (1)

no-treated sample and a pristine foil)

Comment: 

what is the difference between no-treated and pristine foil?

p 3 in sustainability-1656220-peer-review-v1 (1)

On the pristine porous membranes, the contact angle was 98 measured to be 99.4â—¦ , 58.3â—¦ and 0â—¦ for the 0.53µm, 2.4µm and 7.0µm pores, respectively.

Comment: 

explain the change in CA

p 3 in sustainability-1656220-peer-review-v1 (1)

surface energy

Comment: 

there is no surface energy data provided

pp 3 – 4 in sustainability-1656220-peer-review-v1 (1)

So, one can assume that the contact angle 106 (and the associated wettability) of the PET membranes can be (at least partially) controlled 107 by annealing and UV irradiation processes.

Comment: 

explain the reason

p 4 in sustainability-1656220-peer-review-v1 (1)

it is known that reducing pore size leads to increased hydrophobicity

Comment: 

explain the reason

p 4 in sustainability-1656220-peer-review-v1 (1)

For instance, the membranes with 2.4 µm pores show for all 112 treatments a strong hydrophilic behavior - the contact angle is 0â—¦

Comment: 

why it is so?

Reviewer 2 Report

Dear Editor

I carefully reviewed the manuscript entitled “Study of wettability of polyethylene membranes for food packaging”, by Vasi et al., which submitted to Sustainability. The manuscript topic is adequate for the aims and scope of the journal. The authors should address the following comments and revise the manuscript accordingly:

1-Abstract: The abstract has been written in a clear and comprehensive way. Please add one sentence in the initial regarding plastic pollution in environment.

2-There are many papers about the serious plastic environmental pollution. The reference, https://doi.org/10.1016/j.chemosphere.2022.133709, maybe helpful for introduction section, Lines 20-22, and was suggested to be cited.

3- What is the novelty of the research? The necessity and innovation of the article should be presented in the last paragraph of introduction section.

4- Please provide the brands of chemicals and equipment’s used during the experiments throughout the materials and methods section.

5- There are some typographical errors in the paper and authors should be corrected them in the revised version. For example, I think mL is correct.

6- The discussions are very limited. It is suggested to compare the results of the present research with some similar studies which is done before.

7- Please make sure your conclusions' section underscore the scientific value added of your paper, and/or the applicability of your findings/results, as indicated previously. Please revise your conclusion part into more details. Basically, you should enhance your contributions, limitations, underscore the scientific value added of your paper, and/or the applicability of your findings/results and future study in this session.

Reviewer 3 Report

In general the topic has some merit. However, it cannot be published as is. The document lacks references in text as well as in the reference section. In my opinion the results and discussion section should be separated. BOTH sections will then be properly highlighted. 

The conclusion is too long. 

Also a future recommendation section will help 

Author Response

Following the suggestions of the referee the paper was modified. In particular, the results and discussion sections were separated and further developed. Conclusions shortened. Future recommendations were included in the discussion sections. Additional references were included in manuscript.

Reviewer 4 Report

Results section should be improved with more literature and future perspective should be given at conclusion section.

Figure 4 can be formed again clearly. 

Author Response

The authors thank the referee for the suggestions given for the improvement of the manuscript.

In the following our response to the review report:

Results section should be improved with more literature and future perspectives should be given at conclusion section.

R: results section is now improved with more refs, and conclusions are improved.

Figure 4 can be formed again clearly.

R: The fig. 4 was changed to be more clear

Round 2

Reviewer 3 Report

The article entitled "Study of wettability of polyethylene membranes for food packaging" is well written and does is publishable. The only error that must be addressed is the in text citation as well as the reference section in the manuscript. In the first version that was submitted this was the major negative of the paper. 

Author Response

The authors apologize for the error regarding the references. 
The new manuscript version uplaod in .zip file contain a PDF  with all references without errors. 

Bests 
Giovanni Ceccio